# Dynamic Cytoophidia during Late-Stage *Drosophila* Oogenesis

**DOI:** 10.3390/ijms25052575

**Published:** 2024-02-23

**Authors:** Shi-Qi Zhang, Ji-Long Liu

**Affiliations:** 1School of Life Science and Technology, Shanghai Tech University, Shanghai 201210, China; 2Department of Physiology, Anatomy and Genetics, University of Oxford, Oxford OX1 3PT, UK

**Keywords:** CTP synthase, cytoophidium, *Drosophila*, oogenesis, nurse cell, oocyte

## Abstract

CTP synthase (CTPS) catalyzes the final step of de novo synthesis of CTP. CTPS was first discovered to form filamentous structures termed cytoophidia in *Drosophila* ovarian cells. Subsequent studies have shown that cytoophidia are widely present in cells of three life domains. In the *Drosophila* ovary model, our previous studies mainly focused on the early and middle stages, with less involvement in the later stages. In this work, we focus on the later stages of female germline cells in *Drosophila*. We use live-cell imaging to capture the continuous dynamics of cytoophidia in Stages 10–12. We notice the heterogeneity of cytoophidia in the two types of germline cells (nurse cells and oocytes), manifested in significant differences in morphology, distribution, and dynamics. Surprisingly, we also find that neighboring nurse cells in the same egg chamber exhibit multiple dynamic patterns of cytoophidia over time. Although the described dynamics may be influenced by the in vitro incubation conditions, our observation provides an initial understanding of the dynamics of cytoophidia during late-stage *Drosophila* oogenesis.

## 1. Introduction

CTP synthase (CTPS) is the key enzyme catalyzing the final reaction of de novo synthesis of CTP [1]. This reaction transfers the amide nitrogen from glutamine or ammonia to the C-4 position of UTP, thereby generating CTP. GTP has a significant regulatory effect on this reaction. 

In 2010, CTPS was first discovered to form filamentous structures in *Drosophila* ovaries, known as cytoophidia [2]. Cytoophidia exist in the three main cell types of ovarian cells, including follicle cells, nurse cells, and oocytes [2]. Cytoophidia also exist in other tissues of fruit flies, such as the brain, trachea, intestines, male reproductive system, and so on [2]. CTPS forms cytoophidia in a large number of tissues in larvae, especially in the first and second instar stages [3]. Our studies have shown that in third-instar larvae, cytoophidia regulate the tissue architecture and metabolism of fat bodies [4,5]. Cytoophidia have a significant impact on brain development in *Drosophila* larvae [6].

In addition to *Drosophila*, cytoophidia have also been found in other species such as humans [7], yeast [8,9,10], prokaryotes [11], zebrafish [12], plants [13], and archaea [14]. Therefore, the formation of cytoophidia via CTPS is an evolutionarily conserved phenomenon [15].

Using the *Drosophila* ovary as a model, our group and other research groups have conducted a series of studies on the distribution and function of cytoophidia ([3,5,15,16,17,18,19,20,21]). We found that cytoophidia exist in the main cell types of the ovaries, but no significant cytoophidia were observed in two specific types of cells (i.e., stalk cells and polar cells) [22]. Our previous research has focused on the early and middle stages of oogenesis, while we have paid less attention to the late stages of rapid growth and development of the egg chamber. One of the main reasons is that the development of the late-stage egg chamber is particularly fast, making it difficult to capture the dynamics of the late-stage egg chamber in vivo.

Based on these considerations, this study focuses on egg chambers in the later stages, mainly from Stage 10 to Stage 12 (Figure 1). These stages go through the process of nurse cells dumping the main contents into the oocyte. We culture the late-stage egg chamber of CTPS-mCherry transgenic flies in vitro and observe the morphology of the egg chambers and the dynamics of cytoophidia using live imaging, providing us with the opportunity to capture the details of the changes in cytoophidia.

We analyze the images through long-term live imaging and find that (1) CTPS can form cytoophidia in the germline cells in the later stages; (2) the distribution of cytoophidia lies in the nurse cells and oocytes; (3) cytoophidia are dynamic in both nurse cells and oocytes; (4) cytoophidia in nurse cells and oocytes exhibit heterogeneity, manifested in their different shapes, distributions, sizes, and dynamic changes; (5) cytoophidia in individual nurse cells also exhibit different dynamic changes over time. A long-term dynamic analysis of the cytoophidia in the late-stage egg chambers of Drosophila lays the foundation for future understanding of the role of cytoophidia in germline cell development.

## 2. Results

### 2.1. Dynamics of Cytoophidia at Stage 10 Egg Chambers

We performed live imaging of a Stage-10B egg chamber with two channels (Figure 2). The first channel was mCherry labeled CTPS, and the second channel was a bright field image that revealed the contours of two types of germ cells (i.e., nurse cells and oocytes). In the Stage-10B egg chamber, there were different dynamic changes of cytoophidia in nurse cells and the oocyte during the 24-h window period. We have noticed a significant difference in the distribution of cytoophidia between the oocytes and nurse cells. 

Each nurse cell contained one or several relatively large, thick, and long cytoophidia, while the oocyte did not show obvious large cytoophidia. The small cytoophidia in the oocyte were relatively dense. Small cytoophidia in nurse cells were less dense than those in the oocyte. After two hours, the density of small cytoophidia in the oocyte slightly increased, reaching its maximum at the third hour, relatively high at the fourth hour, and gradually decreasing or even disappearing at the fifth hour. At the sixth hour later, there were only small cytoophidia scattered in the oocyte. After the sixth hour, most of the time there were no obvious cytoophidia in the oocyte.

Nurse cells can be divided into four rows based on their distance from the oocyte: the anterior row (farthest from the oocyte) has one nurse cell, followed by the second row with four nurse cells, the third row with six nurse cells, and the fourth row (adjacent to the oocyte) with four larger nurse cells (Figure 3). The first row of nurse cell far from the oocyte had a smaller volume, with a higher density of small cytoophidia. The third and fourth rows tended to have fewer small cytoophidia, but larger cytoophidia appear to be more abundant.

After four hours, the background signal of the nurse cells at the fourth row gradually increased, while the number of large cytoophidia did not show significant changes. For the nurse cell at the first row, the background of cytoophidia gradually increased. Nine hours and ten hours later, we observed an increase in the gap between cells. After 14 h, more cytoophidia were distributed in clusters. In some nurse cells, large cytoophidia gradually disappeared. At the 24th hour, most of cytoophidia in the nurse cell at the first row disappeared. In the second row of nurse cells, there was also a cytoophidium in one of the nurse cells. In the third row, there were nurse cells with cytoophidia, while in the fourth row, there were relatively few cytoophidia.

Through dynamic imaging, we can see that the number of cytoophidia in different nurse cells varies, and there are differences among different cells. This observation indicates that cytoophidia are not uniformly distributed among different nurse cells and exhibit heterogeneity in both time and space. Furthermore, from the perspective of the bright field, the length of the egg chamber during this Stage-10B period increased from one hour to 24 h. There was a slight change in diameter and in volume. The length of the oocyte is also increasing, indicating that the proportion of the oocyte in the entire egg chamber was also increasing.

### 2.2. Dynamics of Cytoophidia at Stage 11/12 Egg Chambers

We further performed live imaging for 24 h of egg chambers from stages 11 to 12 (Figure 4). At the beginning, there were only sporadic cytoophidia in the nurse cells. Over time, nurse cells become smaller and smaller as they dumped most of their contents onto the oocyte. The volume of the oocyte gradually increased and became fuller. After 24 h, the oocyte almost filled the entire egg chamber.

From the mCherry-CTPS channel, mCherry-CTPS was initially relatively uniform in the oocyte. The background of the nurse cells was relatively clean, with some circular-shaped cytoophidia. The background signal of mCherry in the oocyte was relatively high and diffuse, but no obvious cytoophidium was observed. In the later stage, the diffuse color gradually faded away.

A pointed structure could be seen near the head of the egg chamber, which should develop into a dorsal appendage (DA) in the later stage. The DA signal appeared from the sixth hour and gradually lengthened. At the 14th hour, the contrast between DA and the background was very clear. After the 15th hour, there was almost no background, but the DA signal remained present until the 24th hour.

### 2.3. Size Changes of Oocytes and Egg Chambers

We quantified the areas of egg chambers and the ratios of the oocytes to the egg chambers at stage 10B and stage 11/12 (Figure 5). During the 24-h period, the area of the oocyte in the Stage-10B egg chamber gradually increased from 10,000 square micrometers to 15,000 square micrometers (an increase of 50%, Figure 5A). The growth was gradual and increased in a relatively straight line.

The entire egg chamber contains one oocyte and 15 nurse cells. The area of the entire egg chamber was nearly 60,000 square micrometers in the early stages of imaging, and approximately 70,000 square micrometers after 24 h. The increase in egg chamber area was not in a complete straight line, but in a curve. In the first 6 h, the increase in area was linear. From the 7th to 14th hour, there was a slight acceleration in the increase in area. After 14 h, the increase in area began to slow down. From the 18th hour to the 24th hour, there was almost no significant change in the size of the egg chamber, and it was in a plateau period.

For the Stage 11/12 egg chamber, the area of the oocyte increased almost linearly from 21,000 square micrometers to nearly 40,000 square micrometers within 24 h (Figure 5B). The area of the entire egg chamber gradually increased to nearly 50,000 square micrometers. Accelerate the ascent from 5 to 10 h. After 10 h, the gradual platform period was approaching 60,000 square micrometers. In the Stage 11/12 egg chamber, the proportion of oocytes in the entire egg chamber increased significantly. In stages 10–12, although the volume growth rate in oocytes was consistent, the number of cytoophidia in nurse cells decreased precisely when the volume of the egg chamber suddenly increased after reaching its peak (Figure 2 and Figure 4).

We further compared the dynamic changes in the proportion of Stage 10B and 11/12 oocytes in the entire oocyte chamber within 24 h (Figure 5C). In Stage 10B, the proportion of the oocyte slightly increased, from approximately 18% to 20%. In the Stage 11/12 egg chamber, the proportion of the oocyte increased from 45% to 70%. By comparison, the proportion of oocyte to entire egg chamber at Stage 11/12 was much larger than that at Stage 10B.

The proportion of Stage 11/12 oocyte to the egg chamber formed a curve within 24 h, with a significant increase in proportion observed in the first 1 to 7 h. There was a slowing process in the growth of the proportion of the oocyte later on. This slowing process corresponds to the end of the process of nurse cells dumping substances onto the oocyte. The proportion of oocytes remains unchanged for the next 17 h.

### 2.4. Dynamics of Cytoophidia at Stage 10 Nurse Cells

To further pinpoint the dynamics of cytoophidia, we took a close look at three areas of the nurse cells in a Stage 10B egg chamber (Figure 6). For the duration of 15 h, we selected snapshots from the 1st, 6th, and 11th hours. There are three small boxes that mark the three areas of the nurse cells with pink, green, and yellow, respectively.

The pink box marks the second and third rows of nurse cells. The diagram shows the relationship between two adjacent nurse cells (Figure 6a). The large cytoophidium indicated by the purple arrow showed little changes at the 1st and 2nd hours. Dynamic changes occurred at the 3rd hour. Obvious bending occurred at the 4th hour and downward bending did at the 5th hour. 

The cytoophidium indicated by the green arrow is relatively far away from the one indicated by the pink arrow within 1 h (Figure 6a). But over time, these two cytoophidia approached each other. At the 6th hour, the two cytoophidia were relatively close, but there was clearly a gap between them. From bright field, these two cytoophidia resided in two adjacent nurse cells. 

After the 7th and 8th hours, these two cytoophidia changed according to their own trajectory as they approached. We were not sure if it was a coincidence or other factors that brought the two cellular snakes together. But after 9 or 10 h, the two cytoophidia moved in different directions. At the 11th hour, the cytoophidium indicated by a pink arrow began to curl up in another direction. After 15 h, the distribution of the two cytoophidia was almost perpendicular.

The green box in Figure 6b focuses on two cytoophidia, with the purple arrow indicating that the cytoophidium changed from a curved state to a relatively straight state within 5 h. The green arrow refers to a circular shaped cytoophidium. This circular cytoophidium was flipping over.

As time passed, the cytoophidium indicated by the purple arrow rolled, becoming even straighter and thinner. After 6–15 h, the cytoophidia became much thinner than at the beginning. And the circular cytoophidium indicated by the corresponding green arrow was becoming increasingly prominent. At the 10th hour, the cytoophidium appeared to be a closed circle, and at the 11th and 12th hours, it seemed that the cytoophidium was starting to melt.

Figure 6c also shows a nurse cell in the row near the oocyte, with a purple arrow indicating a cytoophidium. At 1 h, the cytoophidium appeared in an O-shape, with another smaller cell snake nearby. In 2 h, this large cytoophidium stretched out very large, looking like a dotted question mark. It stretched at the 3rd and 4th hours. Then, at the 5th hour, a question mark with dots appeared again.

From the 6th to the 11th hour, the cytoophidium continued to maintain a question mark shape. By 11 h, it started to become a bit thinner. At the 13th hour, the cytoophidium suddenly became much shorter. In the 14th hour, only a short stick shape can be seen, and the arc was invisible. At the 15th hour, the cytoophidium disappeared. The cytoophidium exhibited a state of ablation, especially in the last two hours from the 13th to the 15th hour, where it dissolved relatively quickly.

Cytoophidia in three different fields of view and different positions of nurse cells exhibited varying behaviors over time. Some cytoophidia melt relatively quickly. The commonality of these cytoophidia was that they were all in a highly dynamic process.

### 2.5. Hererogeneity of Cytoophidia at Stage 10 Nurse Cells

To appreciate the heterogeneity of cytoophidia, we analyzed the dynamics of cytoophidia within 24 h in Stage 10B nurse cells (Figure 7). In this set of images with hourly intervals, cytoophidia were commonly distributed in different locations of nurse cells at the 1st hour. Cytoophidia exhibited two types in nurse cells: one type was relatively large, with one to several large cytoophidia in each nurse cell (yellow rectangles at the 1:00-h point); another type of cytoophidia was relatively short and small (the green rectangle at the 1:00-h point) (Figure 7).

After 1 h, the number of cytoophidia in nurse cells with fewer cytoophidia sharply decreased, especially in the two anterior rows of nurse cells. Within 3 h, the number of small cytoophidia in the anterior nurse cell became very low; meanwhile, most of the two rows of nurse cells near the oocyte did not have a significant impact. At this point, a clear boundary was formed between the cells based on the background of cytoophidia. At 4 to 5 h, cytoophidia were almost invisible in the 1–3 rows of nurse cells.

At the 5th hour, clear cytoophidia could still be seen in all the nurse cells in the fourth row (Figure 7, the green rectangle at the 5:00-h point) and one or two nurse cells in the third row. Compared to the beginning, the number of cytoophidia tended to decrease.

The fourth row of nurse cells showed no significant changes for several hours, and then from the 13th hour onwards, the background of the cells began to blur. By the 14th hour, the number of cytoophidia was decreasing. By the 18th hour, the background of cytoophidia was almost non-existent, and the larger cytoophidia had almost disappeared. From the 19th to the 24th hour, almost all nurse cells showed no cytoophidia. 

### 2.6. Dynamics of Cytoophidia at Stage 10 Oocytes

To understand the dynamics of cytoophidia in oocytes, we then examined the Stage-10B oocytes closely (Figure 8). In the first example, we analyzed the images every half-hour from 2 h and 30 min to 6 h (Figure 8A). In the case of nurse cells with many cytoophidia, there were not many cellular snakes in oocytes after 2.5 h. Within 3 h, the number of cytoophidia in oocytes gradually increased, and no larger cytoophidia were observed. The background of the small cytoophidia in 3.5 to 4 h was a lot, but it started to decrease by 4.5 h. Within 5 to 6 h, only sporadic cytoophidia were present in the oocyte. There were also a large number of cytoophidia in the same period of nurse cells.

Figure 8B showed the dynamics of the 10B egg chamber from 0.5 h to 4 h. At the beginning, the number of cytoophidia in the oocyte was particularly dense, and there were also many larger cytoophidia that filled the entire oocyte. The state with a large number of cytoophidia was maintained for 2 h. After 2 h, the number of cytoophidia began to decrease but remained relatively thick. By 2.5 h, the background suddenly became sparse, leaving only some thicker cytoophidia. The largest cytoophidia were larger than the 2-h ones, indicating a process of cytoophidium fusion or growth. The background of small cytoophidia changed less, and became lighter after 3 h.

In the anterior side of the oocyte, there was a relatively large number of cytoophidia, resulting in a relatively long entanglement state. At 3.5 h, cytoophidia suddenly disappeared. The appearance of cytoophidia in the oocytes of Figure 8A,B is quite different. In Figure 8A, the number of cytoophidia increased from few to many, then to none. In Figure 8B, cytoophidia gradually disappeared completely from many. These observations suggest that cytoophidia exhibit different dynamic characteristics based on their state.

### 2.7. Dynamics of Cytoophidia at Stage 11/12 Oocytes

We examined two examples of Stage 11/12 egg chambers, with two small time intervals (Figure 9). Figure 9A showed that the background of Stage 11/12 egg chamber cells was not very obvious, with some speckles appearing from 1 h to 7 h, but it is not a clear cell snake. After the 15th hour, there was a slightly blurry signal, but no significant large cytoophidia appeared.

In another example, the background of mCherry is relatively blurry (Figure 9B). After 5 h, the background was slightly brighter. Within 7 h, small dot like structures could be seen, some of which were elongated. After 9 h, the background became relatively clean. At the 11th hour, a dotted signal distribution can be seen on the surface of the egg chamber, but there was relatively little signal inside the oocyte and the background was relatively clean.

## 3. Discussion

This study uses live imaging to record the dynamics of *Drosophila* egg chambers over a long period of time. We find that CTPS can form cytoophidia in germline cells. There are cytoophidia present in both nurse cells and oocytes. And we find that cytoophidia are dynamic in both oocytes and nurse cells. We further observe that the dynamics of cytoophidia in oocytes and nurse cells are different, with heterogeneity manifested in the distribution, dynamics, morphology, and rate of change of cytoophidia. What surprised us is that the dynamic changes of cytoophidia in adjacent nurse cells within the same egg chamber also show significant differences.

### 3.1. Dynamics of Cytoophidia

Under in vitro incubation condition, we observe that cytoophidia are abundant in stage 10 germline cells and become less obvious at stages 11 and 12. In stage 10 nurse cells, cytoophidia can be divided into two categories based on their size: macro-cytoophidia are large sized, having relatively fewer numbers, with one to several in each cell; micro-cytoophidia are small in size, with as many as hundreds per cell [2]. Macro-cytoophidia and micro-cytoophidia seem to be in a dynamic equilibrium. In general, micro-cytoophidia disappear first, and then macro-cytoophidia undergo significant changes. In oocytes, sometimes there are no macro-cytoophidia, only micro-cytoophidia. In stages 11 and 12 egg chambers, macro-cytoophidia are hardly detectable.

In the later stage of the egg chamber, the overall dynamics of cytoophidia are from more to less, which is a process of diassembly. It is highly likely that disassembly begins with a gradual process, but when a certain threshold is reached, cytoophidia will quickly disappear. The rapid appearance of a disassembly window in a short period of time indicates that the dynamic changes of cytoophidia are not a homogeneous and gradual process.

### 3.2. Heterogeneity of Cytoophidium Dynamics

Heterogeneity is manifested on one hand in the differences in the morphology, quantity, and distribution of cytoophidia in oocytes and nurse cells. On the other hand, the distance between the nurse cells and the oocyte in the same egg chamber, or the dynamic changes of adjacent nurse cells, may vary. This is different from the dynamics of cytoophidia in follicle cells [15]. 

In fact, nurse cells and the oocytes are connected through cytoplasmic bridges called ring canals [20]. The cytoplasm of germline cells in the same egg chamber should be interconnected. However, the heterogeneity of cytoophidium dynamics in different cells within the same egg chamber indicates that ring canals can maintain relative cell independence while connecting adjacent cells.

### 3.3. Potential Roles of Cytoophidia

During Stages 10 to 12, the nurse cells dump a large amount of content onto the oocyte, which is a rapidly changing process. Cytoophidia may play multiple roles during late oogenesis. (1) Cytoophidia serve as the reservoir for metabolic enzymes. (2) Metabolic enzymes form cytoophidia, facilitating rapid transportation between cells. (3) Cytoophidia can also serve as a buffer system, and the formation of cytoophidia can maintain a relatively stable range of free enzyme concentrations in cells. (4) Cytoophidia may enhance the efficiency of cells in responding to rapid environmental changes.

### 3.4. Pros and Cons of In Vitro Culture

In order to facilitate the rapid observation of the dynamic changes of cytoophidia in Stages 10 to 12 of the egg chamber, we cultured the egg chambers in vitro. The advantage of doing so is that it can be cultured for a long time to capture the long-term dynamics of cytoophidia. We must also acknowledge that the in vitro culture may not be exactly the same as the in vivo system, and the dynamics of cytoophidia may differ from those at the physiological state within the flies. It is also possible that the disassembly of cytoophidia during late oogenesis that we observed was due to degradation of egg chambers under in vitro incubation condition. For long-term imaging, it would be beneficial to optimize the incubation conditions to mimic the in vivo physiological environment. If technology permits, it is worth observing the dynamics of cytoophidia in vivo. Nevertheless, we believe that the in vitro observations described in this study lay a solid foundation for further understanding the dynamics of cytoophidia in the future.

## 4. Materials and Methods

### 4.1. Fly Strains and Growth Condition

*Drosophila melanogaster* strains were maintained at 25 °C in an incubator, on a standard cornmeal-based medium in vials. The *Drosophila* strain in experiments is CTPS-mCherry transgenic flies, constructed in a laboratory [25].

### 4.2. Ovaries Preparation by Dissection

A group of 5–15 female flies 2–4 days after eclosion, together with several male flies, were transferred into a fly bottle and a wet yeast paste and incubated at 25 °C for about 1–2 days, producing a large number of Stages 10–12 egg chambers.

Intact ovaries were dissected in ~3 mL Grace’s Insect Medium in 35 mm petri dishes and excess tissue was stripped. The dissected ovaries were transferred into a 35 mm dissecting dish with a 200 uL pipette, and time-lapse photography was performed immediately.

### 4.3. Imaging

A clean 35 mm glass-bottom dissecting dish was filled with ~3 mL Grace’s Insect Medium and placed into the dissected ovaries. The ovaries were gently separated into individual egg chambers with the back end of the dissecting tweezers and slowly transferred to the center of the culture hole at the bottom of the dissecting dish.

We employed wide-field illumination (bright field or fluorescence) to identify suitable stage 10–12 egg chambers, positioning them at the center of the dissecting dish before capturing images with a high-power objective lens (40× inverted oil lens). Utilizing the time-lapse function of the Nikon confocal microscope with four dimensions (XYZT), we carefully selected optimal imaging parameters. These included a resolution of 1024 × 1024 pixels, a time interval of 15 min, and a total duration of 24 h. To maintain precision during the process, it was essential to avoid rapid acceleration when switching between egg chambers to prevent any inadvertent movement. This approach ensured the acquisition of high-quality images for analysis.

### 4.4. Statistical Analysis

We opened 4D (x, y, z, t) movies in Fiji ImageJ and exported images at different points in time as needed. Each image was reconstructed to 3D in Imaris and adjusted to the appropriate fluorescence intensity and contrast. Then we used these processed images to rotate, crop, add scale bars, and perform other operations.

When quantifying the egg chamber and oocyte area, we circled the outer outline in Photoshop with the magnetic frame selection tool, and then filled in the interior with a distinct color (such as red). Finally, we used Python (Version 3.10) to identify the pixel with the color RGB, so as to calculate the area. The data were polynomial fitted with Origin (Version 2021).

## 5. Conclusions

Cytoophidium assembly, as a new form of primary partitioning, is increasingly recognized. Its conservatism, universality, and diversity of functions are described in different contexts. In this article, we find that during the late stage of oogenesis in fruit flies, cytoophidia exhibit dynamics. Moreover, these dynamics exhibit heterogeneity across different cell types and at different locations within the same cell type, demonstrating the dynamic, reversible, and robust nature of cytoophidium assembly and disassembly. This work records the dynamics of cytoophidia in vitro for a long time, providing a reference for further understanding the dynamics and functions of cytoophidia in vivo and other environments in the future.

## Figures and Tables

**Figure 1 ijms-25-02575-f001:**
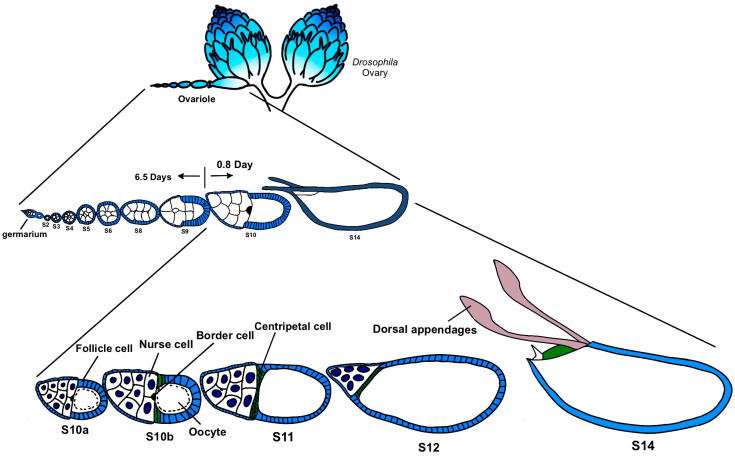
***Drosophila* oogenesis.** The development of *Drosophila* egg chambers can be divided into 14 stages. Early stages include stage 1 in germarium to stage 6 (S6) and middle stages includes Stages 7–9. Late stages span from Stage 10a (S10a) to Stage 14 (S14). This study focuses on Stages 10–12, a rapid progression of oogenesis. Modified from [23,24].

**Figure 2 ijms-25-02575-f002:**
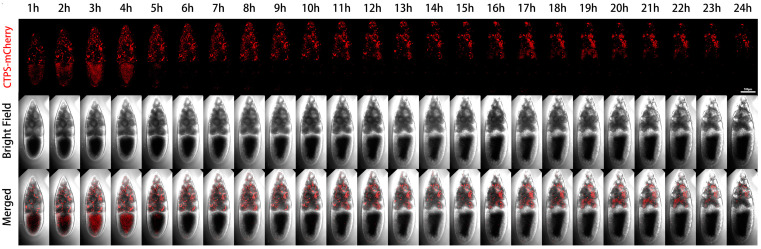
**Dynamics of cytoophidia at Stage 10 egg chambers.** Time-lapse confocal images of Stage 10 *Drosophila* egg chambers expressing indicated proteins. Time 1 h marks the time point when the egg chambers developed for 1 h under microscope. Scale bar, 100 μm. CTPS is tagged with mCherry.

**Figure 3 ijms-25-02575-f003:**
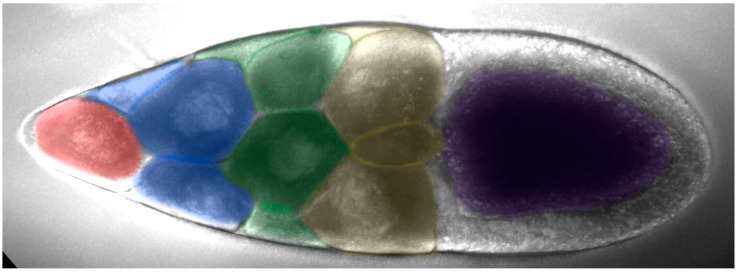
**The arrangement of nurse cells in a *Drosophila melanogaster* egg chamber.** Nurse cells can be divided into four rows based on their distance from the oocyte (purple-colored): the anterior row (pink-colored, farthest from the oocyte,) has one nurse cell, followed by the second row with four nurse cells (blue-colored), the third row with six nurse cells (green-colored), and the fourth row (yellow-colored, adjacent to the oocyte) with four larger nurse cells.

**Figure 4 ijms-25-02575-f004:**
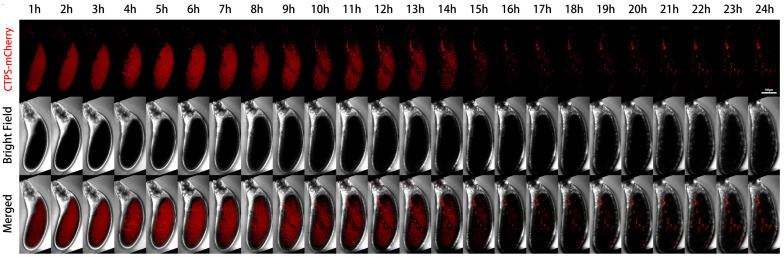
**Dynamics of cytoophidia at Stage 11/12 egg chambers.** Time-lapse confocal images of Stage 11/12 *Drosophila* egg chambers expressing indicated proteins. Time 1 h marks the time point when the egg chambers developed for 1 h under microscope. Scale bar, 100 μm. CTPS is tagged with mCherry.

**Figure 5 ijms-25-02575-f005:**
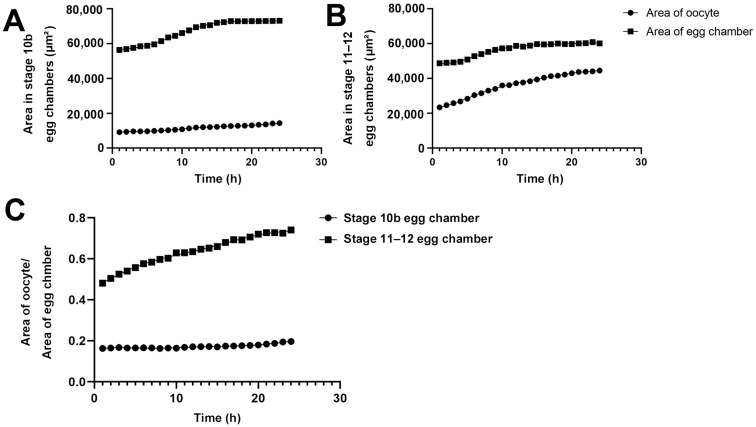
**Size changes of oocytes and egg chambers.** (**A**) Quantification of dynamic changes of area of Stage 10b oocyte and egg chamber. (**B**) Quantification of dynamic changes of area of Stage 11–12 oocyte and egg chamber. (**C**) Quantification of the relative ratio of area of oocyte to area of the whole egg chamber. All data is derived from measurements of Figure 1 and Figure 2.

**Figure 6 ijms-25-02575-f006:**
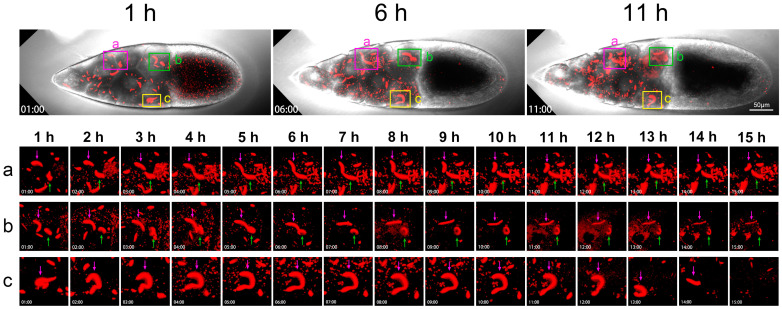
**Dynamics of cytoophidia at Stage 10 nurse cells.** Time-lapse confocal images of a Stage 10a *Drosophila* egg chamber. In the nurse cell region, the cytoophidia in three regions (**a**–**c**) are selected for zoom in, showing their changes in 15 h (1–15 h). Main structures are marked with pink and green arrows. Scare bar, 50 μm. CTPS is tagged with mCherry.

**Figure 7 ijms-25-02575-f007:**
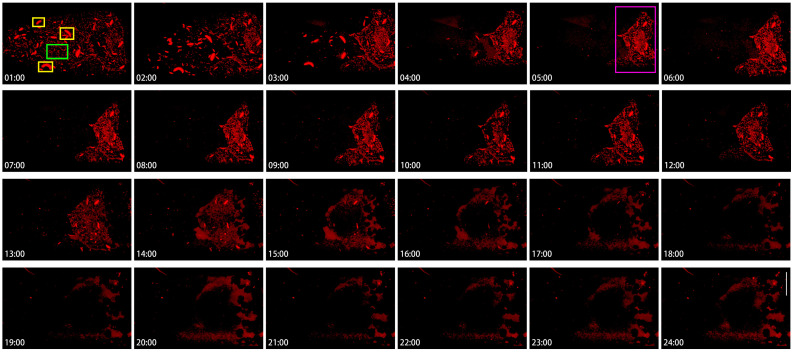
**Hererogeneity of cytoophidia at Stage 10 nurse cells.** Time-lapse confocal images of the Stage 10b nurse cell region in a *Drosophila* egg chamber. Changes in cytoophidia is over a 24-h period. Scare bar, 50 μm. CTPS is tagged with mCherry. Cytoophidia exhibited two types in nurse cells: one type was relatively large, with one to several large cytoophidia in each nurse cell (yellow rectangles at the 1:00-h point); another type of cytoophidia was relatively short and small (the green rectangle at the 1:00-h point). At the 5th hour, clear cytoophidia could still be seen in all the nurse cells in the fourth row (the green rectangle at the 5:00-h point).

**Figure 8 ijms-25-02575-f008:**
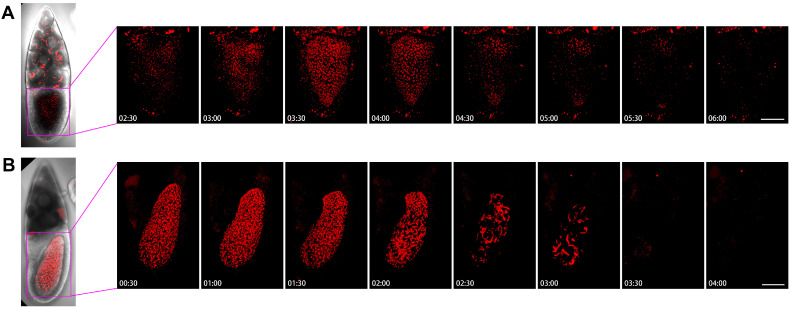
**Dynamics of cytoophidia at Stage 10 oocytes.** Time-lapse confocal images of oocyte region in Drosophila egg chambers. (**A**,**B**) are at Stage 10, the interval is 30 min. Scare bar, 50 μm. CTPS is tagged with mCherry.

**Figure 9 ijms-25-02575-f009:**
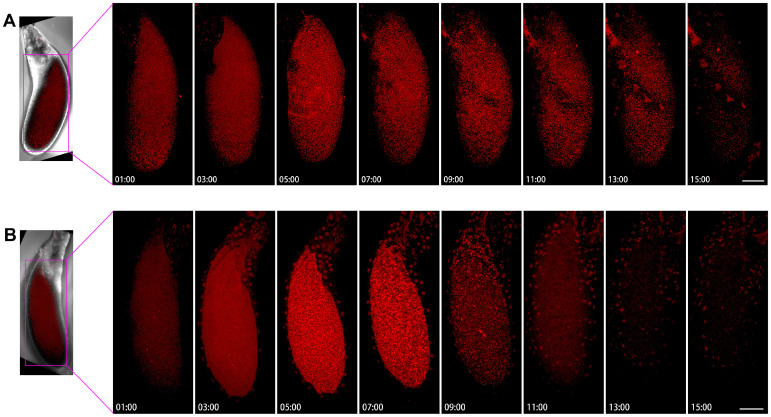
**Dynamics of cytoophidia at Stage 11/12 oocytes.** Time-lapse confocal images of oocyte region in Drosophila egg chambers. (**A**,**B**) are at stage 11/12, the interval is 2 h. Scare bar, 50 μm. CTPS is tagged with mCherry.

## Data Availability

All data is available in the main text.

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
