# Peer review of "Dynamic Cytoophidia during Late-Stage *Drosophila* Oogenesis"

_ijms, 2024, doi:10.3390/ijms25052575_

Round 1
Reviewer 1 Report
Comments and Suggestions for Authors
This study employs live imaging to investigate the dynamics of Drosophila egg chambers over an extended period. The findings reveal the formation of cytoophidia in germline cells, present in both nurse cells and oocytes. Notably, the dynamics of cytoophidia differ between oocytes and nurse cells, exhibiting heterogeneity in distribution, morphology, and rate of change. The study categorizes cytoophidia into macro- and micro-cytoophidia, observing a dynamic equilibrium between them. The research highlights a process of disassembly in the later stages of the egg chamber, suggesting non-homogeneous and rapid changes. Additionally, the study explores the heterogeneity of cytoophidium dynamics within the same egg chamber, revealing potential roles and proposing in vitro culture as a valuable tool for observing long-term dynamics. While acknowledging differences from in vivo systems, the in vitro observations lay a foundation for future investigations.
At this point I have some concerns and suggestion for authors to improve the manuscript.
Figure 4: To enhance the clarity of the figure, consider incorporating hours on the X-axis to provide a more detailed temporal context.
Figure 5 and lines 252-255: To enhance reader convenience, it is advisable for the authors to distinguish these elements in figures by using colored arrows and provide corresponding labels in the accompanying text.
Figure 7: Improving the visualization of information can be achieved by systematically assigning numbers and labels to images, coupled with clear citations placed adjacent to relevant lines in the results text. This approach ensures a more organized and comprehensible presentation. For instance, incorporating Figure numbers and corresponding citations directly into the results text enhances the reader's ability to correlate textual information with visual representations.
By adopting this practice, readers can easily navigate and understand the visual evidence supporting the study's findings. This strategy enhances the overall coherence of the presentation, making it more accessible and facilitating a seamless integration of visual and textual elements. The utilization of numbered and labeled images, coupled with precise citations, not only streamlines the comprehension process but also contributes to a more effective communication of research outcomes.
Further the discussion of results in the present study appears limited, lacking a comprehensive integration with relevant literature as citations are notably absent. Instead, the discussion section reads more like an extension of the results, missing the critical context provided by previous research. To enhance the scholarly value of the paper, the authors should incorporate relevant citations, drawing connections between their findings and established knowledge in the field.
Additionally, the manuscript would benefit from a robust conclusion section that succinctly summarizes the key findings and their implications. A well-crafted conclusion should elucidate the significance of the study, highlighting its contributions to the existing body of knowledge. By explicitly stating how this exploration will benefit scientists and readers in the future, the authors can provide a clearer roadmap for potential applications and further research directions. Including a thoughtful conclusion will not only enhance the completeness of the manuscript but also contribute to the broader scientific discourse by contextualizing the study's relevance and potential impact.
Comments on the Quality of English Language
The manuscript requires revision for English language proficiency, particularly in the method section where numerous sentences are composed in a directive manner.
For example, lines 82-86 "Use wide field to illuminate (bright field or fluorescence) and find suitable stage 10-12 egg chambers, locate them in the center of the dissecting dish and photographed with a high-power objective lens, here using a 40x inverted oil lens. Use the Nikon confocal microscope's time-lapse function with 4 dimensions XYZT, and select the appropriate im-aging parameters. Here, the resolution is 1024*1024 pixels, the time interval is 15 minutes, and the duration is 24 hours. The acceleration when switching egg chambers should not be too fast to prevent the egg chambers from moving."
Can be revised as "We employed wide-field illumination (bright field or fluorescence) to identify suitable stage 10-12 egg chambers, positioning them at the center of the dissecting dish before capturing images with a high-power objective lens (40x inverted oil lens). Utilizing the time-lapse function of the Nikon confocal microscope with four dimensions (XYZT), we carefully selected optimal imaging parameters. These included a resolution of 1024*1024 pixels, a time interval of 15 minutes, and a total duration of 24 hours. To maintain precision during the process, it was essential to avoid rapid acceleration when switching between egg chambers to prevent any inadvertent movement. This approach ensured the acquisition of high-quality images for analysis."
By adopting a more descriptive and grammatically correct structure, the revised sentences enhance the clarity and professionalism of the manuscript. It is advisable to apply similar revisions throughout the method section to ensure consistency and improved readability.
Reviewer 2 Report
Comments and Suggestions for Authors
The manuscript by Zhang and Liu describes the distribution of the enzyme CTP synthase during stages 10 to 12 of oogenesis in Drosophila females. For this the author use a previously established and published line with CTPS tagged with Cherry from which egg chambers are kept over many hours in vitro allowing microscopy over an extended period of time. Beyond its function in metabolism, the interesting aspect about CTPS is its aggregation into cytophidia, which have been described previously for several species, cell types including oogenesis in Drosophila. In the current manuscript the authors take a close look at a specific stage during oogenesis. Figure 1 gives and overview of 24 hours in stage 10, figure 2 of the following late stage. Figure 3 provides some numbers for the growth of wild type egg chambers, figure 4 provides images and image series in high magnification of single cytophidia. Figure 5 contains images showing accumulation of cytophidia in the nurse cells covering many hours of stage 10 (similar to figure 1). Figure 6 provides images about the temporary appearance of cytophidia in the oocyte, althouth with larger images than in figure 1. Figure 7 provides images of late stage egg chambers similar to figure 2 although with larger images.
The manuscript is purely descriptive. No hypothesis or scientific question about CTPS or cytophidia is raised. Only wild type genotypes are observed. The data remain anecdotal, since the dynamics of the cytophidia is not quantified in any way, no tracking of particle movements, shape changes or growth/shrinkage. The image data consist of single movies with one hour frame rates. The only quantification is shown in Figure 3 which is a measurement of the areas of egg chamber and oocyte from the movie over 24 hours. It is not clear to me, what is novel about this and what is the link to cytophidia and CTPS. The data are from a single movie with no averaging and estivation of variance.
I am in favor of descriptive reports, if they provide accessible, useful and novel data. The manuscript in its current version does not provide any of those criteria. Since it would be useful to have a reference to the dynamics of CTPS cytophidia in oocytes, I would recommend to condense the manuscript. Figure 1 may provide an overview of cytophidia in stages 10 to 12. Figure 2 and Figure 3 may focus on specific aspects, which would include a few, 3 to 5 snap shots in high magnification on individual particles showing their changes over time. Accompanying those figures should be any sort of quantification such as intensity changes, growth, shape change, movement, fusion/fission events.
Reviewer 3 Report
Comments and Suggestions for Authors
Zhang and Liu explored the dynamics of cytoophidia during Drosophila oogenesis through long-term live imaging. Cytoophidia are filamentous structures formed by CTP synthase (CTPS). The authors employed mCherry-tagged CTPS, observing dynamic changes in cytoophidia formation during late oogenesis. They characterized the heterogeneity and dynamics of cytoophidia. While their findings are intriguing and contribute to a deeper understanding of the potential involvement of CTPS in oogenesis, certain points should be addressed to meet the standards of IJMS.
Major points
- The authors have acknowledged the cons and limitations of long-term live imaging in the discussion, with a major concern being the compatibility of CTPS signals with in vivo conditions. I would point out that submerging samples in media without additional supplements, especially oxygen and optimal nutrients, could induce severe stress conditions, potentially altering metabolisms and gene expressions. It is crucial for the authors to validate that their samples develop normally without experiencing severe stress during the extended incubation period. Given that stage 10 is generally shorter than 24 hours, the authors can provide evidence, such as the observation of follicle contraction or border cell migration (stages 9-10), during the extended incubation time, which would serve to validate the condition of the tissues.
- Additionally, the authors should address whether they observed all the patterns of cytoophidia from freshly dissected and fixed ovaries. Theoretically, the ovary should contain a wide spectrum of different developmental states in terms of timing. This observation would support their claim that the findings from the system are consistent with in vivo conditions.
- Overall, the decreasing signal intensity and areas over time in mid or late stages raise the question of whether this could be due to protein degradation. It would be beneficial for the authors to consider and discuss this possibility, especially given the unique characteristics of oocytes lacking transcription but maintaining active translation and degradation processes. To address this concern comprehensively, observations of similar patterns from freshly dissected and fixed samples (see above) would provide additional support and eliminate potential confounding factors.
- It is interesting that CTPS displays dynamic patterns over time with distinct features. While the authors provide a few representative images, a more comprehensive characterization and quantification with additional details (such as the number of samples, percentages of linear, bending, '?' shape, etc., at specific time points, like 1 hour, 6 hour, 12 hour..) would enhance the clarity and depth of their findings.
Minor points
- I suggest adding higher magnification images and mark each part of egg chamber for the readers who are not familiar with flies.
- Also mark or add lines to illustrate the four dividing nurse cells.
- Moreover, the authors' quantification of oocyte size (Fig 3) is already well-characterized. To enhance the informativeness of their findings, they might consider correlating the changes in CTPS patterns with oocyte size. This correlation could provide valuable insights into the relationship between CTPS dynamics and oogenesis progression.
Round 2
Reviewer 1 Report
Comments and Suggestions for Authors
I do not have any additional questions or suggestions regarding the manuscript, except to recommend that authors who did not consider adding a conclusion section during the initial revision should do so now. A well-defined conclusion section would be beneficial, as it would effectively connect the study findings to broader aspects of knowledge. I recommend accepting the manuscript after this revision is made.
Reviewer 3 Report
Comments and Suggestions for Authors
The authors addressed my comments and modified the manuscripts accordingly.
